# Genetic Insights into the Extremely Dwarf *Hibiscus syriacus* var. *micranthus*: Complete Chloroplast Genome Analysis and Development of a Novel dCAPS Marker

**Soon-Ho Kwon [1], Hae-Yun Kwon [2] and Hanna Shin [1,\*]**

[1] Department of Forest Bio-Resources, National Institute of Forest Science, Suwon 16631, Republic of Korea; shkwon84@korea.kr

[2] Forest Medicinal Resources Research Center, National Institute of Forest Science, Yeongju 36040, Republic of Korea; kwonhy05@korea.kr

\* Correspondence: hanna193@korea.kr; Tel.: +82-31-290-1165

**Abstract:** This study explored the chloroplast (cp) genomes of three *Hibiscus syriacus* (HS) specimens endemic to Korea possessing unique ornamental and conservation values: the dwarf *H. syriacus* var. *micranthus* (HSVM), renowned for its small stature and breeding potential; HS 'Tamra', a cultivar from Korea's southernmost islands, noteworthy for its distinctive beauty; and HS Natural Monument no. 521 (N.M.521), a specimen of significant lifespan and height. Given the scarcity of evolutionary studies on these specimens, we assembled and analyzed their cp genomes. We successfully assembled genomes spanning 160,000 to 160,100 bp and identified intraspecific variants. Among these, a unique ATA 3-mer insertion in the *trnL-UAA* region was identified in HSVM, highlighting its value as a genetic resource. Leveraging this finding, we developed a novel InDel dCAPS marker, which was validated across 43 cultivars, enhancing our ability to distinguish HSVM and its derivatives from other HS cultivars. Phylogenetic analysis involving 23 Malvaceae species revealed that HSVM forms a clade with woody *Hibiscus* species, closely associating with N.M.520, which may suggest a shared ancestry or parallel evolutionary paths. This investigation advances our understanding of the genetic diversity in Korean HS and offers robust tools for accurate cultivar identification, aiding conservation and breeding efforts.

**Keywords:** *Hibiscus syriacus* var. *micranthus*; chloroplast genome; dCAPS marker; *trnL-UAA* p8 region; dwarf

## 1. Introduction

*Hibiscus syriacus* L. (HS), commonly known as the national flower of Korea, is a deciduous shrub belonging to the Malvaceae family [1]. HS is native to Korea and southern China, and during its approximately 100-day summer bloom, its flowers are displayed in a diverse array of colors, including white, pink, red, blue, and purple, and various shapes, including single, semi-double, and double forms [2–4]. These distinctive characteristics have led to the development of numerous cultivars with ornamental value worldwide [5]. In recent years, there has been an increasing interest in the breeding of dwarf types of HS suitable for indoor cultivation in Korea. A variety of methods has been used to develop these cultivars, such as the induction of mutations by irradiation (e.g., 'Dasom', 'Ggoma', 'Kids Purple', and 'Kids White'), intraspecific crossing with dwarf materials (e.g., 'Saehanseo' and 'Red Bohanjae'), and graft-induced phenotypic variation (e.g., 'Andong') [6–12].

*Hibiscus syriacus* var. *micranthus* Y. N. Lee & K. B. Yim (HSVM) is a natural dwarf variety of HS identified in Andong, Korea in 1992 (Gyeongsangbuk-do Tangible Cultural Property No. 28) [13]. Characterized by its small and erect structure with unbranched growth, it was only 1.2 m in clear length of trunk even at approximately 100 years old when discovered. The flower of HSVM has five whirled white petals and each petal is adorned

with distinct pink-blue eye spots that do not overlap. The petal is 1.8 cm in length and 6 mm in width, and the pistil is 2.3 cm long (Figure 1a). In particular, HS 'Andong' is a grafting mutant of HSVM that retains the peculiar flower traits of its progenitor while exhibiting improved growth [14]. It has been utilized to develop many dwarf cultivars, such as HS 'Simbaek', 'Cheoyong', and 'Chungam' [15,16]. Aside from reports on the phenotypic traits of this variant and its use in breeding programs, there is currently limited information on the origin, evolutionary relationships, and genetic characteristics of HSVM, contributing to a gap in our understanding of this variety.

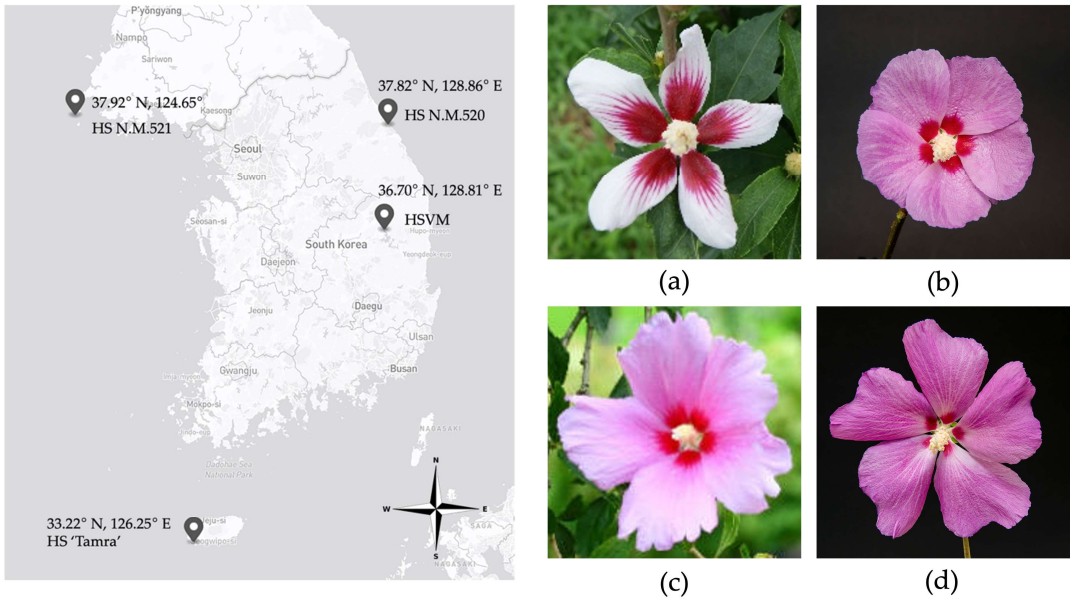

**Figure 1.** Geographic discovery sites of *Hibiscus syriacus* (HS) specimens (**left**) and floral morphology (**right**). (**a**) HSVM; (**b**) HS N.M.521; (**c**) HS N.M.520; (**d**) HS 'Tamra.' HSVM: *Hibiscus syriacus* var. *micranthus*; HS N.M.521: HS Natural Monument no. 521; HS N.M.520: HS Natural Monument no. 520; HS 'Tamra': *Hibiscus syriacus* cultivar 'Tamra'.

As an autonomous entity within plant cells, the chloroplast (cp) is essential for energy conversion and photosynthesis, and also for its potential in plant genetics and breeding [17–19]. The unique properties of chloroplast DNA (cpDNA), such as its maternal inheritance and lack of recombination, make it a reliable marker for plant taxonomy and phylogenetics. Given its relatively small genome size and the presence of highly conserved genes, cpDNA facilitates the precise identification of plant species and their evolutionary paths [20]. The maternal inheritance pattern of cpDNA is particularly useful for studying population dynamics and phylogeography, offering insights into the migratory routes and historical distribution of plant species. In addition, the conserved sequence regions of cpDNA, along with variable intergenic spacers, provide a rich source of polymorphic markers necessary for resolving phylogenetic relationships at various taxonomic levels [21].

In this study, we explored the genetic properties of HSVM by assembling and comparing its cp genomes with those of other HS trees with extended lifespan in Korea. Specifically, we assembled the complete cp genomes of HSVM, renowned HS Natural Monument no. 521 (HS N.M.521), and the cultivar HS 'Tamra' from Jeju Island. We conducted an in-depth comparative analysis of their cp genome sequences, including the already known cp genome of HS Natural Monument no. 520 (HS N.M.520) [22]. Using phylogenetic analysis, we attempted to determine the evolutionary position of HSVM by including various closely related species. Further, we identified intraspecific cp variations in HSVM, leading to the development of a specialized derived cleaved amplified polymorphic sequence (dCAPS) marker that can be used to distinguish between cultivars [23].

## 2. Materials and Methods

### 2.1. Specimen Collection and Conservation for cp Genome Assembly

In this study, cp genomes were assembled from three specimens—HSVM, HS N.M.521, and HS 'Tamra'—selectively collected from different regions of Korea, namely Andong, Gyeongsangbuk-do (36.70° N, 128.81° E); Baengnyeongdo Island (37.92° N, 124.65° E), the westernmost point of central Korea; and Jeju Island (33.22° N, 126.25° E), the southernmost point of Korea, respectively (Figure 1). These specimens were preserved in the *Hibiscus* Clonal Archive at the National Institute of Forest Science, located in Suwon, Korea (37.15° N, 126.57° E). HSVM is a rare natural dwarf tree in *Hibiscus* spp. worldwide. In contrast, HS N.M.521 is characterized by purple flowers with red eye spots and stands at a height of approximately 6.3 m. The HS 'Tamra' features large purple flowers with very small and faint eye spots.

### 2.2. DNA Extraction, Sequencing, Assembly, and Annotation

For cp genome sequencing, fresh leaf samples were collected from the three HS specimens. Total DNA extraction was performed using the GeneAll® Exgene™ Genomic DNA Purification Kit (GeneAll Biotechnology, Seoul, Republic of Korea). The next-generation sequencing library was prepared with Macrogen (Seoul, Republic of Korea), using the TruSeq Nano DNA Kit (Illumina, San Diego, CA, USA), and the sequencing was conducted on the HiSeq 2500 platform (Illumina Inc., San Diego, CA, USA). The cp genomes were assembled using NOVOPlasty v.4.3.1, which was configured with k-mers of 27, 31, and 33 to optimize the de Bruijn graph complexity for various genome regions [24]. The annotation and circular map construction were carried out with the GeSeq web tool https://chlorobox.mpimp-golm.mpg.de/geseq.html (accessed on 3 January 2022), using blatN and blatX annotators, in conjunction with Chlorom v0.1.0 [25].

### 2.3. Comparative Analyses of cp Genome Sequences

The sequence alignments necessary for pinpointing variations within the cp genomes of the HS specimens were executed using Clustal Omega v.1.2.4 [26]. Pairwise comparison analyses were performed to detect gaps, differences, and sequence identities, with a threshold of 99% for sequence identity and a minimum of a 10 base pair gap for variant differentiation using CLC Main Workbench software v23.0.2 [27]. To comprehensively compare the cp genome sequences, we used the mVISTA program https://genome.lbl.gov/vista/index.shtml (accessed on 11 January 2024) [28]. Individual-specific variants were identified if supported by a minimum of five reads with a base quality score of $\geq 30$ using default parameters. The analysis was conducted by employing GATK's HaplotypeCaller version 4.2.4 and the SeqIO module from Biopython version 1.8 [29,30]. Moreover, for the validation of variant calls, Sanger sequencing data were aligned using the AlignX tool integrated in Vector NTI Advanced version 10.3.0 [31].

### 2.4. Phylogenetic Analysis

The primary objective of the phylogenetic analysis was to ascertain the evolutionary position of HSVM in the Malvaceae family. To this end, we included a total of 23 species in the analysis, comprising 13 species of the genus *Hibiscus*, 4 species of the genus *Abelmoschus*, and 5 species of the genus *Gossypium*. *Tilia amurensis* was strategically selected as the outgroup to serve as the root for the phylogenetic tree. We focused on 78 conserved coding sequences (CDS) from the cp genome, which are critical for resolving phylogenetic relationships in this family [32]. Sequence alignment was executed using Clustal Omega version 1.2.4 with default parameters, ensuring precise comparison across all species. The optimal phylogenetic model was found by using the best-fit model (TVM + F + I + G4) of ModelFinder version 2 [33], with the Bayesian information criterion implemented in IQ-TREE version 2.2.6 [34]. According to the best-fit model, the maximum likelihood tree was constructed with 1000 bootstrap replicates using IQ-TREE2. The tree was visualized

using CLC Main Workbench version 23.0.2 to solidify confidence in the phylogenetic node placement.

### 2.5. InDel dCAPS Marker Design

The dCAPS method was selected for marker development due to its high specificity in detecting single nucleotide polymorphism (SNP) or short insertion/deletion (InDel) variations, which is instrumental for precise genetic mapping in populations with low genetic diversity [35]. The dCAPS Finder version 2.0 http://helix.wustl.edu/dcaps/ (accessed on 10 May 2023) facilitated the design of primers that introduce a restriction site in the presence of a target SNP, enabling the use of restriction enzymes for allele discrimination. Primer efficacy was verified using Oligoevaluator http://www.oligoevaluator.com/LoginServlet (accessed on 10 May 2023), which assessed parameters such as melting temperature, self-complementarity, and secondary structure potential to ensure high amplification efficiency [36]. The selection of suitable restriction enzymes for the dCAPS assays was performed using Enzymefinder ver. 2.13.1 http://enzymefinder.neb.com/ (accessed on 1 June 2023), considering factors such as enzyme sensitivity to DNA methylation and star activity.

### 2.6. Validation of dCAPS Marker across Diverse HS Cultivars

For dCAPS marker validation, fresh leaf samples from 40 HS specimens were collected. The cultivars were HS 'Andong', 'Baekdanshim', 'Byeollee', 'Byeonghwa', 'Cheoyong', 'Chungam', 'Hairi', 'Hanbit', 'Hanmaum', 'Hanoltanshim', 'Hanyang', 'Hwahap', 'Ggoma', 'Oknyo', 'Saebit', 'Saehan', 'Simbaek', 'Simsan', 'Sondok', 'Soyang', 'Suni', 'Taewha', 'Tanshim', 'Wonwha', 'Yaum' developed in Korea, 'Akagionmamori', 'Blue Bird', 'Campanha', 'Dorothy Crane', 'Heikeyama', 'Helene', 'Hinomaru', 'Ishigakijima', 'Lil Kim', 'Mostrosus', 'Pheasant Eye', 'Shigyoku', 'Shiroshorin', 'Wood Bridge' in foreign cultivars, and HS N.M.520. The DNA was extracted using the method described in Section 2.2. To perform polymerase chain reaction (PCR), 1 μL of extracted HS genomic DNA (10 ng), 2 μL of dCAPS primer set (0.01 μM of each forward and reverse primer), 0.25 μL of TaKaRa Taq™ DNA Polymerase (Takara Bio Inc., Kusatsu, Japan) (5 U/μL), 2 μL of 10× PCR buffer, 4 μL of dNTP Mixture (2.5 mM each), and 10.75 μL of distilled water were mixed to prepare a total of 20 μL of reaction mixture. The process consisted of an initial denaturation at 94 °C for 5 min; followed by 36 cycles of denaturation at 94 °C for 30 s, annealing at 53 °C for 30 s, and extension at 72 °C for 1 min; with a final extension at 72 °C for 10 min. The PCR product was then subjected to a reaction with 0.4 μL of *Mlu*CI restriction enzyme, 1 μL of buffer, and 3.6 μL of distilled water for a total of 10 μL at 37 °C for 2 h. The digested PCR product was subsequently run on a 2% TAE agarose gel at 100 V for 80 min.

## 3. Results

### 3.1. Genome Assembly and Summary

We successfully assembled the complete cp genomes for the three *Hibiscus* individuals using the sequencing data. For HSVM, a total of 74,438,950 reads were generated, with 2,626,042 aligned reads and 1,828,976 assembled reads, constituting 3.53% of the organelle genome, with an average organelle coverage of 2463×. For HS N.M.521, a total of 78,191,088 reads were recorded, with 3,149,562 aligned reads and 3,136,530 assembled reads, representing 4.03% of the organelle genome and an average organelle coverage of 2953×. Lastly, in HS 'Tamra', the total reads were 75,767,768, with 2,701,392 aligned reads and 2,490,624 assembled reads, comprising 3.57% of the organelle genome and achieving an average organelle coverage of 2535×. The complete circular cp genomes of HSVM, HS N.M.521, and HS 'Tamra' were 161,022 base pair (bp), 161,027 bp, and 160,899 bp, respectively (Figure 2). The accession numbers for the three individuals were deposited in the NCBI GenBank under OM_687473, OM_687472, and OM_541594, respectively. When comparing the cp genomes of four individuals including the results for HS N.M.520, whose cp genome had already been assembled, we observed that HS 'Tamra' had the smallest total genome

size at 160,899 bp, followed by HS N.M.520, HSVM, and HS N.M.521 in ascending order. The cp genomes of all four individuals contained 130 genes, including 85 protein-coding, 37 transfer RNA (tRNA), and eight rRNA genes (Table 1).

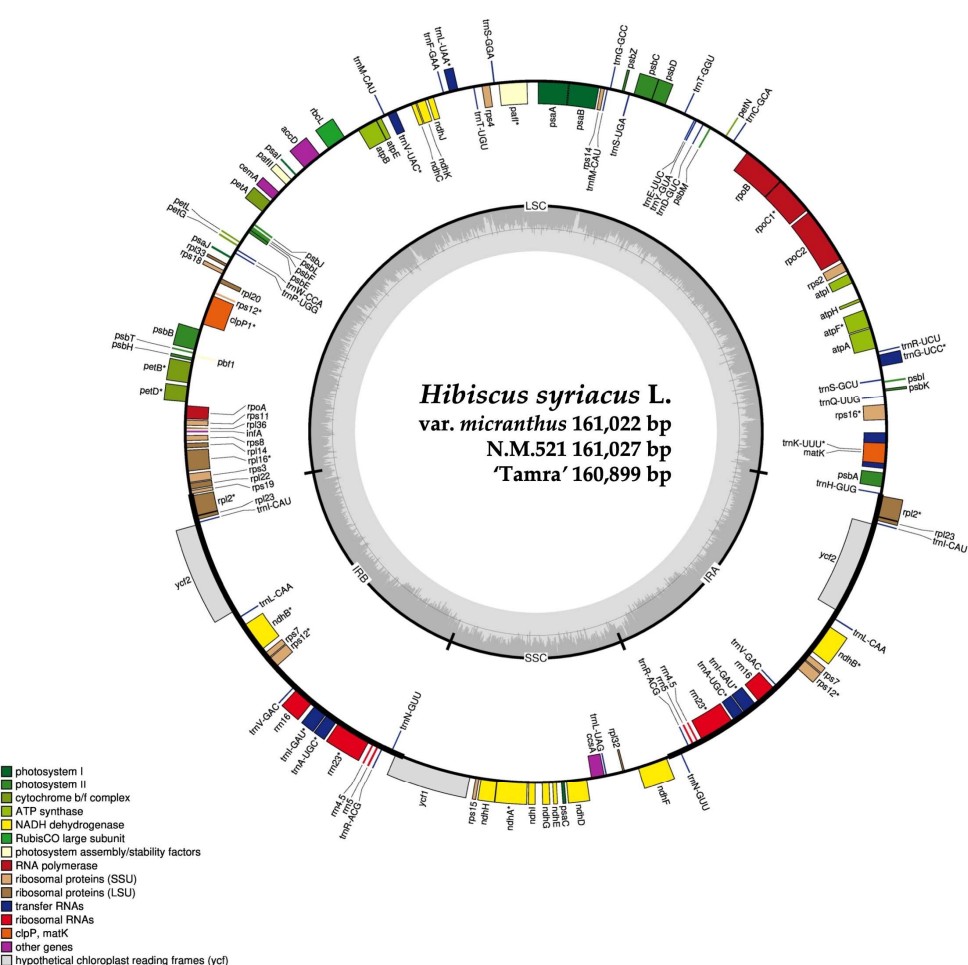

**Figure 2.** Circular map of the chloroplast genome of three *Hibiscus syriacus* specimens. Outer circle depicts genes, tRNAs, and rRNAs with different colors. Inner circle highlights the quadrant structure, with dark gray indicating GC content. LSC (large single-copy), IR (inverted repeat), and SSC (small single-copy) regions are marked. The asterisk indicates intron-containing gene.

**Table 1.** Summary of the complete chloroplast genomes of three *Hibiscus syriacus* specimens.

| Name | Genome Size (bp) | LSC | IRB | SSC | IRA | Number of Genes | Protein-Coding Genes | tRNA | rRNA | GC Contents (%) |
|---|---|---|---|---|---|---|---|---|---|---|
| HSVM | 161,022 | 89,701 | 25,745 | 19,831 | 25,745 | 130 | 85 | 37 | 8 | 36.83 |
| HS N.M.521 | 161,027 | 89,706 | 25,745 | 19,831 | 25,745 | 130 | 85 | 37 | 8 | 36.83 |
| HS 'Tamra' | 160,899 | 89,755 | 25,742 | 19,660 | 25,742 | 130 | 85 | 37 | 8 | 36.84 |

HSVM: *H. syriacus* var. *micranthus*, HS N.M.521; *H. syriacus* Natural Monument no. 521; HS 'Tamra': *H. syriacus* 'Tamra'.

### 3.2. Gene Annotation and Variations in Genes

A total of 130 genes in all specimens were categorized into four groups, namely photosynthesis, transcription, translation, and others (Table 2). Within the photosynthetic gene cluster, genes related to photosystems I and II, the cytochrome b6f complex, ATP synthase, and NADPH dehydrogenase were identified. Notable variations were observed in *psbC*, *ndhA*, and *atpF*, which had specific mutations such as insertions, deletions, and substitutions in the different specimens. For example, each *psbC* and *ndhA* gene exhibited

an insertion in the HS 'Tamra' specimen. In addition, the *psbC* gene in HS 'Tamra' also had a substitution, while *atpF* displayed a deletion in the HSVM variant. Genes associated with transcription and translation, small and large subunits of ribosomes, initiation factors, and various RNA components were included.

**Table 2.** Gene contents in the cp genome of HSVM.

| Role | Group of Gene | Name of Gene | No. |
|---|---|---|---|
| Photosynthesis | Photosystem I | *psaA, psaB, pasC, psaI, psaJ* | 5 |
| | Photosystem II | *psbA, psbK, psbI, psbM, psbD, psbF, psbC* [4]*, psbH, psbJ, psbL, psbE, psbN, psbB* | 13 |
| | Cytochrome b/f complex | *psbT, psbZ, petN, petA, petL, petG, petD* [1]*, petB* [1] | 8 |
| | ATP synthase | *atpI, atpH, atpA, atpF* [1,4]*, atpE, atpB* | 6 |
| | Cytochrome c-type synthesis | *ccsA* | 1 |
| | Assembly/stability of photosystem I | *ycf3 (pafI)* [3]*, ycf4 (pafII)* | 2 |
| | NADPH dehydrogenase | *ndhB* \*,[1]*, ndhH, ndhA* [1,4]*, ndhI, ndhG, ndhJ, ndhE, ndhF, ndhC, ndhK* [4]*, ndhD* | 12 |
| | Rubisco | *rbcL* | 1 |
| Transcription and translation | Small subunit of ribosome | *rpoA, rpoC2, rpoC1* [1,4]*, rpoB, rps16* [1]*, rps2, rps14, rps4, rps18* [4]*, rps12* \*\*\*,[1]*, rps11, rps8, rps3, rps19, rps7* \**, rps15* | 18 |
| | Large subunit of ribosome | *rpl33, rpl20, rpl36, rpl14, rpl16* [1]*, rpl22, rpl2* \*,[1]*, rpl23* \**, rpl32* | 11 |
| | Translational initiation factor | *infA* | 1 |
| | Ribosomal RNA | *rrn16* \**, rrn4.5* \**, rrn5* \**, rrn23* \* | 8 |
| | Transfer RNA | *trnH-GUG, trnK-UUU* [1,4]*, trnQ-UUG, trnS-GCU, trnS-UCC* [1]*, trnR-UCU, trnC-GCA, trnD-GUC, trnY-GUA, trnE-UUC* [1],\*\**, trnI-GGU, trnS-UGA, trnG-UCC, trnfM-CAU* \*\**, trnS-CGA* [4]*, trnT-UGU, trnL-UAA* [1,4]*, trnF-GAA, trnV-UAC* [1]*, trnW-CCA, trnP-GGU, trnL-CAA* \**, trnV-GAC* \**, trnA-UGC* [1],\**, trnR-ACG* \**, trnN-GUU* \**, trnL-UAG, trnI-CAU* | 37 |
| Others | RNA processing | *matK* [4] | 1 |
| | Carbon metabolism | *cemA* | 1 |
| | Fatty acid synthesis | *accD* | 1 |
| | Proteolysis | *clpP* [2,4] | 1 |
| | Component of TIC complex | *ycf1* [4] | 1 |
| | Hypothetical proteins | *ycf2* \*,[4] | 2 |
| Total number of genes | | | 130 |

[1] Contains one intron within the gene. [2] Contains two introns. [3] Contains three introns. [4] Contains variants. \* Gene duplicated within the cp genome. \*\* Gene triplicated within the cp genome. \*\*\* Gene with trans-spliced exons.

Among the 85 protein-coding genes (Table 1), 12 were consistently present in the cp genome of all HS specimens: *petD*, *petB*, *atpF*, *ycf3*, *ndhB*, *ndhA*, *rpoC1*, *rps16*, *rps12*, *rpl16*, *rpl2*, and *clpP*. Except for *ycf3*, which had three introns, and *clpP*, which had two introns, the genes contained a single intron. Among the 37 tRNAs, eight were *trnK-UUU*, *trnS-UCC*, *trnL-UAA*, and *trnV-UAC*, with two copies of *trnE-UUC* and two copies of *trnA-UGC*, each containing one intron. Genes in this category showed insertions in *rpoC2* and a substitution in *rps18* specific to HS 'Tamra' and HS N.M.521, respectively. The tRNA genes essential for protein synthesis, such as *trnK-UUU* and *trnS-GCU*, exhibited substitutions in the HS N.M.521 and HSVM samples. The HS 'Tamra' specimen exhibited a deletion in the *trnS-GGA* gene. Other genes, such as *matK* and *cemA*, involved in RNA processing, carbon metabolism, and proteolysis showed substitutions across specimens. The HS 'Tamra' variant exhibited a unique pattern of gene variation, including insertions and substitutions not observed in the other specimens. Overall, the variation in GC content was low, with HS 'Tamra' showing a slightly higher percentage than the others, suggesting a potential impact on genomic stability and functionality (Table 2).

### 3.3. Comparative Analysis of Genome Structure and Sequence Variability

Comparative analysis of the cp genome structures of the four HS specimens revealed notable size variations. The HSVM specimen presented minor size differences across its genome compared with others. Specifically, its LSC region was 89,701 bp, differing slightly from HS N.M.520 by +3 bp and from HS N.M.521 by −5 bp. Moreover, HS N.M.521 had the largest LSC at 89,706 bp. The SSC region of HSVM measured 19,831 bp, consistent with

the sizes of HS N.M.520 and HS N.M.521. All three specimens shared identical IR regions, measuring 25,745 bp. In contrast, the 'Tamra' specimen displayed significant variations, with a 3-bp reduction in the IR region, the LSC region extended by 49 to 57 bp, and a smaller SSC region measuring 19,660 bp.

Analysis of the marginal regions of quadripartite cp genome structures showed that the positioning of the genes at the boundaries slightly varied among these individuals. The *rps19* gene straddled the boundary of the LSC region by 3 bp in all of the HS specimens, and the *rpl2* gene was situated 114 bp away from the boundary. At the end of the IRb region, the position of the *ycf1* gene, which is situated at the junction between the IRb and SSC regions, coincided precisely with the boundary in the HSVM and HS 'Tamra' specimens, with the terminal stop codon aligning with the border. In contrast, HS N.M.521 and HS N.M.520 had the *ycf1* gene extending beyond the boundary by 2 bp. In the SSC region, the gene *rps15* was situated at a significant distance from the border with the IRb region. For HSVM, HS N.M.520, and HS N.M.521, this gene was positioned 5498 bp away from the junction. However, HS 'Tamra' displayed a variation in that its *rps15* gene started 6 bp closer to the boundary, compared with the other HS specimens. As for the boundary between the SSC and IRa regions, the *ycf1* gene, which extends across this junction, overlapped by 698 bp in HSVM, HS N.M.520, and HS N.M.521. In HS 'Tamra', the overlap was slightly larger, with the *ycf1* gene encroaching 3 bp further into the IRa region, compared with the other specimens. At the interface between the IRa and the LSC regions, no variation in gene positioning was identified (Figure 3).

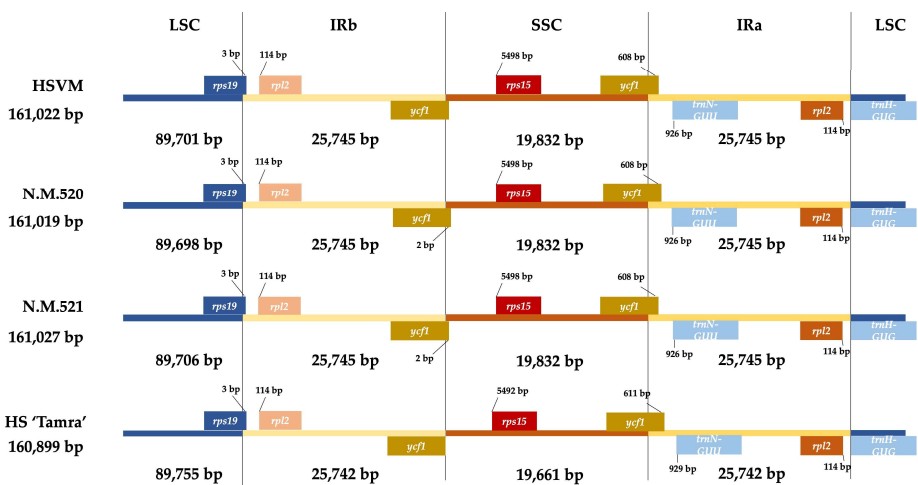

**Figure 3.** Distance between adjacent genes and junctions of the SSC, LSC, and two IR regions among the plastid genomes of four *Hibiscus syriacus* specimens. HSVM: *H. syriacus* var. *micranthus*; HS N.M.520: *H. syriacus* Natural Monument no. 520; HS N.M.521: *H. syriacus* Natural Monument no. 521; HS 'Tamra': *H. syriacus* 'Tamra'. LSC: large single-copy; IRb: inverted repeat b; SSC: small single-copy; IRa: inverted repeat a.

In the comparative sequence variability study of HS, HSVM demonstrated both notable genetic similarities and differences when compared with other specimens. HSVM and HS N.M.520 showed an exceptionally high genetic match, with 99.98% sequence identity. Similarly, HSVM and HS N.M.521 shared a very close genetic relationship with a 99.99% match. These high percentages of identity indicate a strong genetic linkage between these specimens. However, HS 'Tamra' presented a notable contrast, displaying slightly less genetic similarity to HSVM, with a 99.71% match. This lower percentage suggests that HS 'Tamra' has a greater genetic distance from HSVM and the other specimens.

Comparative visualization of the chloroplast genomes across the three HS specimens using mVISTA showed that the variability is primarily concentrated in certain genomic regions. Notably, HS 'Tamra' exhibited a higher frequency of individual-specific variations, particularly in the non-coding regions and to a lesser extent in the *trnL-CAA*, *rps16*, *trnQ-*

*UUG*, and *psbM* genes. In contrast, the HSVM and HS N.M.521 specimens displayed more conserved genomic segments with fewer variations (Figure 4).

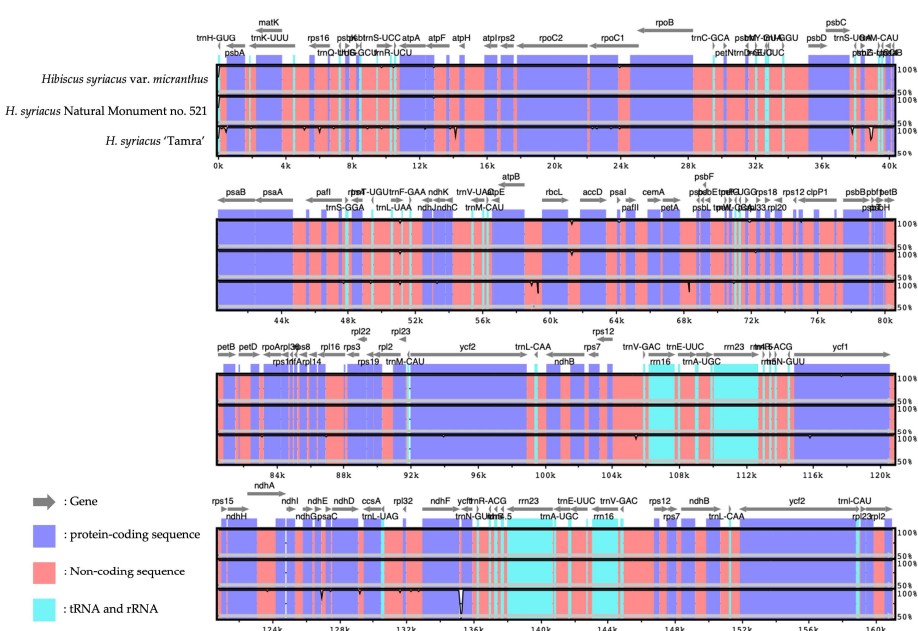

**Figure 4.** Visualization of alignment identity between four HS. Alignment analysis was conducted using the Shuffle-LAGAN method. Sequences were annotated and identified using different colors. Sequence identity ratio is presented through vertical depth, using *H. syriacus* Natural Monument no. 520 as a reference.

With regard to the number of genetic disparities, HS 'Tamra' exhibited significant differences, with 486 gaps and 468 SNPs, when compared with HSVM. This was a clear indication of its genetic uniqueness. Conversely, the limited number of gaps and SNPs between HSVM and HS N.M.521, specifically 20 gaps and 12 SNPs, reinforced their genetic closeness (Figure 5). Utilizing GATK's HaplotypeCaller and the SeqIO module from Biopython for individual-specific variant analysis, HS 'Tamra' stood out with 44 substitutions, 150 insertions, and 268 deletions. HSVM is characterized by a single substitution and three insertions, displaying minimal but potentially significant variation [37]. HS N.M.520 showed a slightly higher variation with two substitutions and five insertions, along with 10 unique deletions. In contrast, HS N.M.521 displayed the least variation, with only one insertion observed (Table S1).

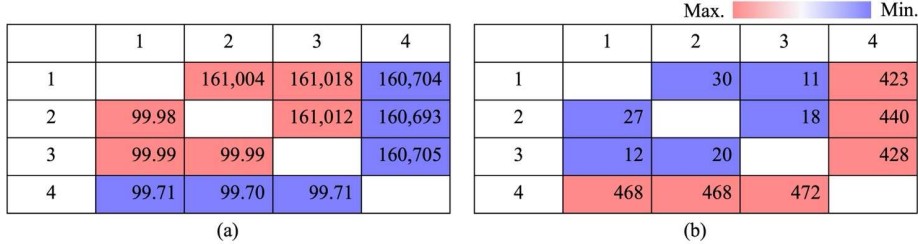

**Figure 5.** Pairwise comparison heatmap. (**a**) Percent identities (bp) and identity percentages (%); (**b**) Gaps and differences of whole genomes; **upper** panel shows gaps (bp), and **lower** panel shows differences (bp). 1: *Hibiscus syriacus* var. *micranthus*; 2: *H. syriacus* Natural Monument no. 520; 3: *H. syriacus* Natural Monument no. 521; 4: *H. syriacus* 'Tamra'.

### 3.4. Phylogenetic Analysis

The phylogenetic analysis conducted in this study revealed valuable insights into the genetic relationships between 23 species in the Malvaceae family. Utilizing a set of 78 CDS,

the phylogram shows that most Korean *Hibiscus* individuals are closely grouped. HSVM demonstrated a clear monophyletic relationship with HS N.M.520, signifying a close genetic connection. This association indicates that they may have a shared ancestry or potentially similar ecological niche or evolutionary history despite their different phenotypes. Notably, HS 'Tamra' is positioned closer to HS 'Purpureus variegatus' and *H. sinosyriacus* than to HSVM or HS N.M.520. HS 'Tamra' was selected from the geographically isolated Jeju Island, and its evolutionary trajectory is seemingly different from that of HSVM.

*T. amurensis*, which also belongs to the Malvaceae family, was set as an outgroup in the phylogenetic tree owing to its evolutionary and taxonomic differences from other genera in the family, such as *Hibiscus*, *Gossypium*, and *Abelmoschus*. It diverged at an early stage and was hence used as a root for the phylogenetic tree [38]. Subsequently, a majority of the woody *Hibiscus* species, including HSVM, formed a monophyletic group together with certain other *Hibiscus* species, encompassing woody and perennial herbaceous types, as well as the genus *Abelmoschus*. Within this grouping, the woody HS were well differentiated from *H. rosa-sinensis*. In the remaining groups, perennial herbaceous species of the *Hibiscus* genus, such as *H. cannabinus* and *H. sabdariffa*, diverged earlier in a paraphyletic relation with other species, whereas the herbaceous species of *Hibiscus* and *Abelmoschus* represented some of the most recently diverged plants (Figure 6).

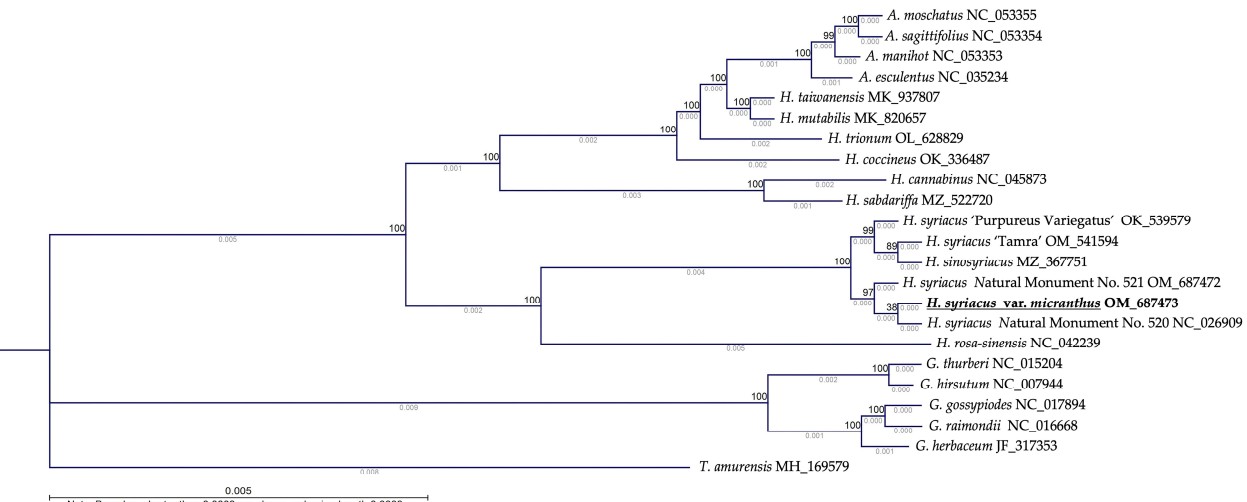

**Figure 6.** Phylogenetic analysis of 23 species in the Malvaceae family based on chloroplast genomes. Phylogram constructed from 78 conserved coding sequences using the maximum likelihood method. The analyses were executed with IQ-Tree ver. 2.2.6 and CLC Main Workbench Version 23.0.2, with bootstrap validation performed 1000 times to establish node confidence.

### 3.5. Development dCAPS Marker Using a Unique Insertion in HSVM trnL-UAA

In the variant analysis, a unique InDel mutation was pinpointed in the *trnL-UAA* region of HSVM. To facilitate the identification of HSVM and cultivars with HSVM as a maternal parent and to preserve the genetic sovereignty of HSVM, we developed a dCAPS marker. A detailed inspection of HSVM's *trnL-UAA* region revealed the insertion of a 3-mer ATA sequence. This insertion occurred in a Group I intron that required external guanosine triphosphate for splicing, particularly in the P8 region, noted for its intron variability (Figure 7) [37]. To utilize this sequence as a genetic marker, primers were crafted to facilitate the recognition of restriction enzymes (Table 3). The enzyme *Mlu*CI, known to cleave blunt ends at AATT sites, was selected for the process. The terminal base of the forward primer was altered from A to C to prevent enzyme recognition, while the last 3′ base of the reverse primer was switched from A to T, allowing cleavage in HSVM. After restriction digestion, this primer set produced bands of 105 bp, 28 bp, and 4 bp in HSVM and its related cultivars due to enzyme activity. In contrast, HS specimens without the ATA insertion yielded a single 134 bp band (Figure 8).

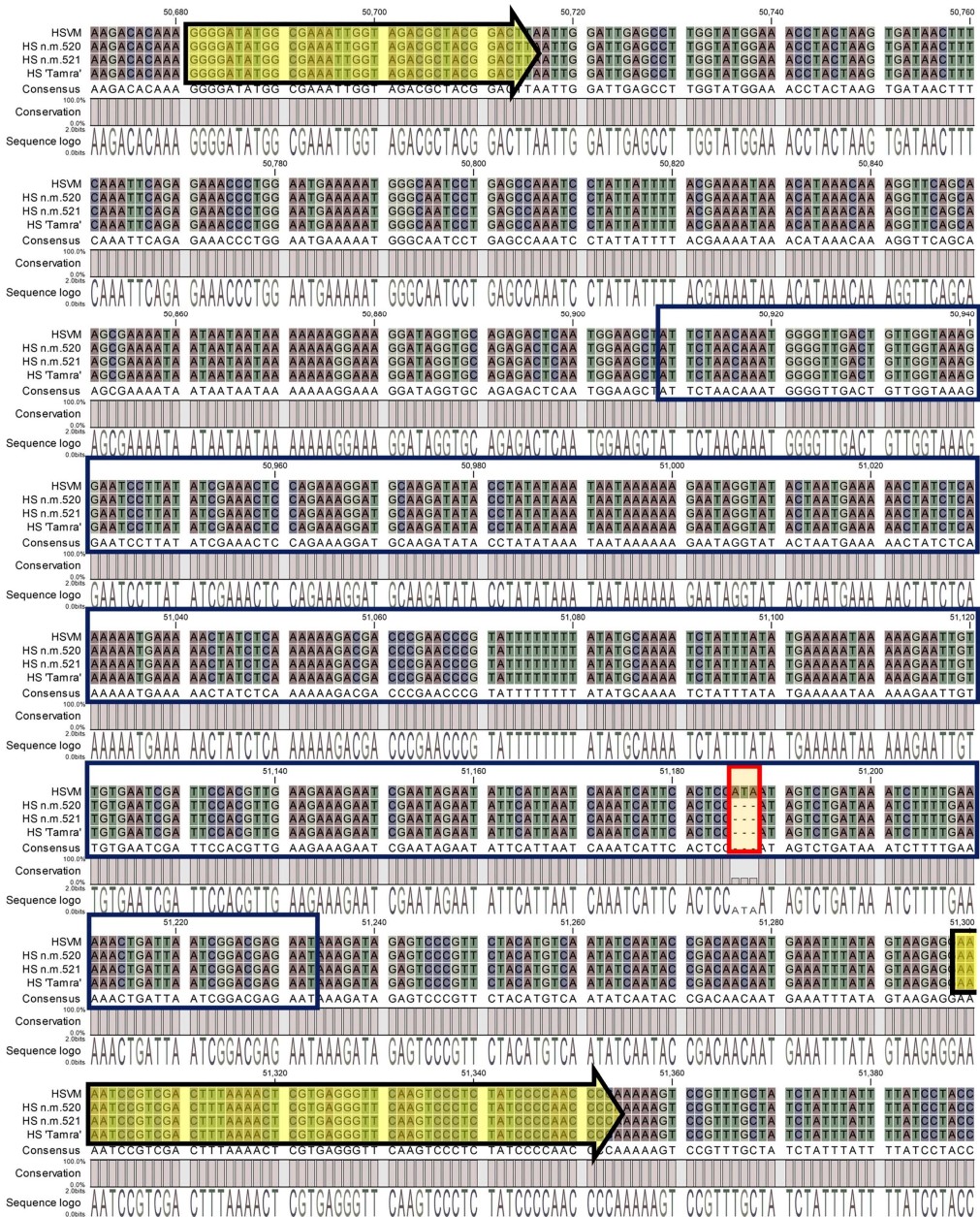

**Figure 7.** Alignment results of the *trnL-UAA* region in four *Hibiscus syriacus*. The yellow arrows point to the tRNA coding regions; the dark blue boxes indicate the P8 area in the intron; and the red box highlights the InDel region. HSVM: *Hibiscus syriacus* var. *micranthus*; HS N.M.520: HS Natural Monument no. 520; HS N.M.521: HS Natural Monument no. 521; HS 'Tamra': *Hibiscus syriacus* cultivar 'Tamra'.

**Table 3.** Sequences of dCAPS marker primers.

| Primer Name | Sequence (5′ → 3′) |
|---|---|
| Hs_dCAPS_F | ATGCAAAATCTATTTATATGAAAAATAAAAAGC |
| Hs_dCAPS_R | AATCAGTTTTTCAAAAGATTTATCAGACA |

(a)  [ATGCAAAATCTATTTATATGAAAAATAAAAAGA→C]ATTGTTGTGAATCG
ATTCCACGTTGAAGAAAGAATCGAATAGAATATTCATTAATCAAATCATTC
ACTCCAT↑AAT[T↓←AGTCTGATAAATCTTTTGAAAAACTGATT]

(b)  [ATGCAAAATCTATTTATATGAAAAATAAAAAGA→C]ATTGTTGTGAATCG
ATTCCACGTTGAAGAAAGAATCGAATAGAATATTCATTAATCAAATCATTC
ACTCC---AT[T←AGTCTGATAAATCTTTTGAAAAACTGATT]

[▢] : primer     ▮ : modified base     [▢] : Enzyme site

**Figure 8.** Design of InDel dCAPS marker. (**a**) The PCR product structure from the HSVM; (**b**) The PCR products structure from a common HS. Square brackets '[]' indicate primer binding regions, the green box marks the enzyme recognition site, and the red highlights denote bases modified to create or disrupt cleavage sites. Yellow highlights indicated reigions in primer that are unmodified.

### 3.6. Discrimination Test of HSVM and Maternal Lines Using InDel dCAPS Marker

DNA samples from HSVM, HS N.M.520, HS N.M.521, and the other 40 HS cultivars were used as PCR templates, followed by restriction enzyme treatment and electrophoresis of the PCR products. We found that 11 cultivars with the ATA insertion, namely HSVM, HS 'Andong', 'Byeonghwa', 'Cheoyong', 'Chungam', 'Hairi', 'Hwahap', 'Lil Kim', 'Simbaek', 'Taewha', and 'Yaum', showed cleaved fragments of 105 bp and 28 bp. In contrast, the PCR products from the remaining HS cultivars were not cleaved, resulting in 134 bp bands (Figure 9). Subsequent bidirectional sequencing of the PCR products revealed the exclusive presence of a 3-bp ATA insertion in 11 cultivars (Table 4). HS 'Andong' has blooming white flowers with red eye spots selected from grafting variants of HSVM, and it is exported under the name HS 'Lil Kim' in the United States. Notably, HS cultivars 'Byeonghwa', 'Cheoyong', 'Chungam', 'Hairi', 'Hwahap', and 'Yaum' were bred with HS 'Andong' as a parent, while HS 'Simbaek' and 'Taewha' were developed using 'Hwahap' and 'Cheoyong' as maternal parents, respectively. This dCAPS marker was able to accurately distinguish not only HSVM but also cultivars resulting from crosses with HSVM as the maternal line or those selected from the subsequent generations derived from these cultivars (Figure 9).

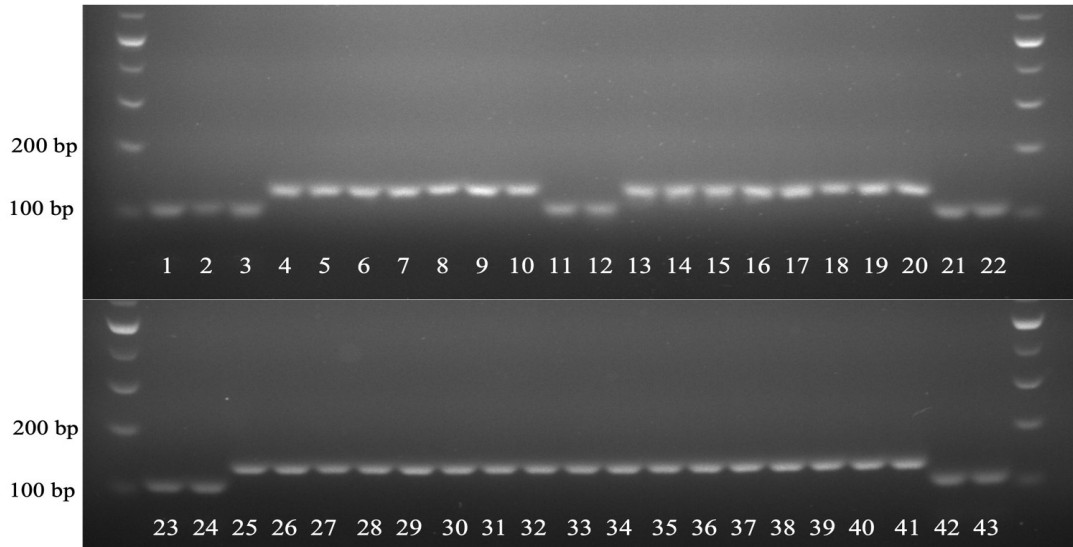

**Figure 9.** Gel electrophoresis using the dCAPS marker. The PCR products were resolved on a 2% TAE agarose gel at 100 V for 80 min. Sample numbering corresponds directly to that listed in Table 4.

**Table 4.** List of *Hibiscus syriacus* cultivars and corresponding *trnL-UAA* InDel sequences.

| No. | Name | Development History | Flower Color | Country | Sequence (5′ → 3′) |
|---|---|---|---|---|---|
| 1 * | HSVM | Regional variety | White with spot | South Korea | CACTCCATA**ATA**GTCTGATA |
| 2 * | HS 'Andong' | HSVM graft mutation | White with spot | South Korea | CACTCCATA**ATA**GTCTGATA |
| 3 * | HS 'Lil Kim' | HSVM graft mutation | White with spot | South Korea | CACTCCATA**ATA**GTCTGATA |
| 4 | HS 'Suni' | HS 'Helene' × HS 'Asadal' | White with spot | South Korea | CACTCCATA---GTCTGATA |
| 5 | HS 'Simsan' | HS 'Helene' × HS 'Mauve Queen' | White with spot | South Korea | CACTCCATA---GTCTGATA |
| 6 | HS 'Wonwha' | Selection breeding | White with spot | South Korea | CACTCCATA---GTCTGATA |
| 7 | HS 'Ggoma' | Radiation breeding | White with spot | South Korea | CACTCCATA---GTCTGATA |
| 8 | HS 'Tanshim' | Selection breeding | White with spot | South Korea | CACTCCATA---GTCTGATA |
| 9 | HS 'Baekdanshim' | Selection breeding | White with spot | South Korea | CACTCCATA---GTCTGATA |
| 10 | HS 'Byeollee' | HS 'Gaeryangdanshim' × (HS 'Baekdanshim' × HSVM) | White with spot | South Korea | CACTCCATA---GTCTGATA |
| 11 * | HS 'Simbaek' | HS 'Hwahap' × HS 'Samchulri' | White with spot | South Korea | CACTCCATA**ATA**GTCTGATA |
| 12 * | HS 'Hairi' | HS 'Andong' × HS 'Hwahap' | White with spot | South Korea | CACTCCATA**ATA**GTCTGATA |
| 13 | HS 'Saebit' | Selection breeding | White with spot | South Korea | CACTCCATA---GTCTGATA |
| 14 | HS 'Sondok' | Selection breeding | White with spot | South Korea | CACTCCATA---GTCTGATA |
| 15 | HS 'Hanmaum' | Selection breeding | White with spot | South Korea | CACTCCATA---GTCTGATA |
| 16 | HS 'Hanbit' | Selection breeding | White with spot | South Korea | CACTCCATA---GTCTGATA |
| 17 | HS 'Hanyang' | Selection breeding | White with spot | South Korea | CACTCCATA---GTCTGATA |
| 18 | HS 'Hanoltanshim' | Selection breeding | White with spot | South Korea | CACTCCATA---GTCTGATA |
| 19 | HS 'Pheasant Eye' | Introduction breeding | White with spot | United states | CACTCCATA---GTCTGATA |
| 20 | HS 'Hinomaru' | Exotic cultivar | White with spot | Japan | CACTCCATA---GTCTGATA |
| 21 * | HS 'Byeonghwa' | HS 'Andong' × HS 'Bulsae' | Purple with spot | South Korea | CACTCCATA**ATA**GTCTGATA |
| 22 * | HS 'Cheoyong' | HS 'Andong' × HS 'Gaeryangjaju1' | Purple with spot | South Korea | CACTCCATA**ATA**GTCTGATA |
| 23 * | HS 'Chungam' | HS 'Andong' × HS 'Gaeryangjaju1' | Purple with spot | South Korea | CACTCCATA**ATA**GTCTGATA |
| 24 * | HS 'Taewha' | HS 'Cheoyong' × HS 'Gwangmyoung' | Purple with spot | South Korea | CACTCCATA**ATA**GTCTGATA |
| 25 | HS N.M. no. 520 | Designated as a natural monument no. 520 | Purple with spot | South Korea | CACTCCATA---GTCTGATA |
| 26 | HS N.M. no. 521 | Designated as a natural monument no. 521 | Purple with spot | South Korea | CACTCCATA---GTCTGATA |
| 27 | HS 'Tamra' | Selection breeding | Purple with spot | South Korea | CACTCCATA---GTCTGATA |
| 28 | HS 'Soyang' | Selection breeding | Purple with spot | South Korea | CACTCCATA---GTCTGATA |
| 29 | HS 'Ishigakijima' | Exotic cultivar | Purple with spot | Japan | CACTCCATA---GTCTGATA |
| 30 | HS 'Heikeyama' | Exotic cultivar | Purple with spot | Japan | CACTCCATA---GTCTGATA |
| 31 | HS 'Campanha' | Exotic cultivar | White with spot | United states | CACTCCATA---GTCTGATA |
| 32 | HS 'Helene' | Exotic cultivar | White with spot | United states | CACTCCATA---GTCTGATA |
| 33 | HS 'Doroshy Crane' | Exotic cultivar | White with spot | England | CACTCCATA---GTCTGATA |
| 34 | HS 'Mostrosus' | Exotic cultivar | White with spot | Belgium | CACTCCATA---GTCTGATA |
| 35 | HS 'Shiroshorin' | Exotic cultivar | White with spot | Japan | CACTCCATA---GTCTGATA |
| 36 | HS 'Blue Bird' | Exotic cultivar | Blue with spot | United states/France | CACTCCATA---GTCTGATA |
| 37 | HS 'Shigyoku' | Exotic cultivar | Blue with spot | Japan | CACTCCATA---GTCTGATA |
| 38 | HS 'Oknyo' | Selection breeding | White without spot | South Korea | CACTCCATA---GTCTGATA |
| 39 | HS 'Saehan' | Selection breeding | White without spot | South Korea | CACTCCATA---GTCTGATA |
| 40 | HS 'Wood Bridge' | Exotic cultivar | Red with spot | England | CACTCCATA---GTCTGATA |
| 41 | HS 'Akagionmamori' | Exotic cultivar | Red with spot | Japan | CACTCCATA---GTCTGATA |
| 42 * | HS 'Yaum' | HS 'Andong' × HS 'Samchulri' | Purple with spot | South Korea | CACTCCATA**ATA**GTCTGATA |
| 43 * | HS 'Hwahap' | HS 'Andong' × HS 'Namwon' | White with spot | South Korea | CACTCCATA**ATA**GTCTGATA |

\* Cultivar derived from HSVM as maternal parent. - Sequence gap.

## 4. Discussion

HSVM is recognized as a unique dwarf variety in the HS group, notable for its distinctive phenotype and exclusive floral traits. This study advances the scientific understanding of its genetic and evolutionary significance. By assembling the cp genome of HSVM and comparing it with those of ancient Korean HS trees and other Malvaceae species, we highlight the dual importance of HSVM, including its intrinsic genetic value and potential as a progenitor in horticultural breeding.

Comparison of our findings with those of previous studies reveals that the cp genome structure of HSVM aligns with the general cp genome organization observed in Malvaceae, with similar variations in genomic size to those reported in other species. However, HSVM exhibits a unique genetic profile, particularly in the *trnL-UAA* region, which has not been observed in other studies. This distinctive InDel mutation provided a foundation for developing a dCAPS marker, instrumental for lineage tracing and maternal parentage verification in breeding programs. This strengthens the practical applications of HSVM's genetic traits.

The phylogenetic placement of HSVM in the Malvaceae family corroborates its close relationship with HS N.M.521, suggestive of shared ancestry and evolutionary trajectories, despite phenotypic variations. Although cp genomes have limitations in identifying phenotype-genotype associations, their utility in evolutionary studies is well established, a theme consistent with previous research findings. Phylogenetic analysis positions *H.*

*sinosyriacus* in the HS clade, suggesting that its current classification may require revision. Future studies are essential to determine its accurate taxonomic status.

Our findings underscore the significance of intronic sequences in cp genomes, particularly the P8 region of Group I introns, for plant diversity and adaptation, aligning with studies that have demonstrated the evolutionary importance of tRNA intron sequences.

This study illuminates the genetic diversity and evolutionary narrative of HS using cp genome analysis of HSVM and Korea-native HS. HSVM is validated as a valuable genetic resource, not just ornamentally but also as a cornerstone for genetic research and breeding. Despite certain constraints of cp genomes in connecting genetic data to phenotypic traits, the evidence of their influence on the evolution of plant genomes and diversity is compelling. Our discovery of the unique role of the P8 region in the cp genome highlights the importance of intronic sequences in plant adaptation and diversity, providing a direction for future research focused on intronic variations in the cp genome and their direct role in plant phenotypic expression.

**Supplementary Materials:** The following supporting information can be downloaded at https://www.mdpi.com/article/10.3390/cimb46030173/s1: Table S1: Individual-specific variant distribution (unit: bp).

**Author Contributions:** Conceptualization, S.-H.K., H.-Y.K. and H.S.; methodology, S.-H.K.; software, S.-H.K.; validation, S.-H.K. and H.-Y.K.; formal analysis, S.-H.K.; investigation, S.-H.K.; resources, H.-Y.K.; data curation, S.-H.K.; writing—original draft preparation, S.-H.K.; writing—review and editing, H.S.; visualization, S.-H.K.; supervision, H.S.; project administration, H.S.; funding acquisition, H.S. All authors have read and agreed to the published version of the manuscript.

**Funding:** This research was funded by the Research Program of the National Institute of Forest Science, grant number FG0403-2023-02-2024.

**Institutional Review Board Statement:** Not applicable.

**Informed Consent Statement:** Not applicable.

**Data Availability Statement:** The original contributions presented in the study are included in the article and Supplementary Materials. Further inquiries can be directed to the corresponding author.

**Conflicts of Interest:** The authors declare no conflicts of interest.

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
