# Peer review of "Genetic Insights into the Extremely Dwarf Hibiscus syriacus var. micranthus: Complete Chloroplast Genome Analysis and Development of a Novel dCAPS Marker"

_cimb, doi:10.3390/cimb46030173_

Round 1

Reviewer 1 Report

Comments and Suggestions for Authors

I have completed my assessment of your manuscript, titled " Genetic Insights Into the Extremely Dwarf Hibiscus syriacus var. micranthus: Complete Chloroplast Genome Analysis and Development of a Novel dCAPS Marker " and would like to provide feedback on the various issues that need attention and revision before publication.

  1. Line 13-14: Why compare with several indigenous plants, please explain clear.
  2. Abstract: The progress of our target plants should be added, and what is the significance of what we are now doing.
  3. Abstract: Regarding the chromatographic fingerprints, I did not see any sort of phylogenetic analysis of these data.
  4. Line 49-52: "there is few information on the origin, evolutionary relationships, and genetic traits",Please make clear.
  5. Line 49-52: He comparison with previous work should be moved to the discussion section.
  6. Line 58-64: Revise these sentences for better scientific clarity.
  7. Line 110-122: There is no need to list so many login numbers.
  8. Line 118: Change"37 transfer RNA (tRNA), and eight rRNA genes." to "37 tRNAs and 9 rRNAs"
  9. Line 180-183: There is no significant difference among the genomes, so no need for a detailed explanation.
  10. Line 190-192: Figure caption: Simplify the figure caption
  11. Line 194: Change"Among the 85 protein-coding genes,." to "Among the 85 protein-coding genes, (Table 1)"
  12. Line 225-236: Avoid unnecessary explanations. Simplify the discussion regarding the number of repeats found in the studied genomes and variations among species.
  13. Line 289.290: Clarify the meaning of"T. amurensis, used as an outgroup in the phylogenetic study, is distinctly 289 placed on the tree, indicative of its early divergence from the Malvaceae lineage".
  14. Line 289-293: Clarify the sentence for better comprehension.
  15. "Table 4." takes up a lot of space.
  16. Section titled "Phylogenetic analysis". Can the authors provide a sense of how many genera there are in Malvaceae, and how many of these had plastomes available?
  17. Line 369-370: The sentence is not clear; please rephrase it for clarity.
  18. Reorder the presentation of results: Begin with protein coding gene length, total gene numbers, GC%, and then discuss protein coding genes.
  19. The Discussion section needs improvement; provide comparisons with previous research.
  20. The Conclusion section is too detailed. It should offer a concise summary of the study's outcomes.
  21. Consider comparing the current cp genome with more related cp genomes to enhance the depth of the research.
  22. Start with a brief description like "Phylogenetic relationships of Malvaceae based on chloroplast genomes". Whether in the first description or the following sentence, mention relationships before genetic distances, as relationships are the most important aspect of the tree. Adding, like "The scale bar indicates".

Comments on the Quality of English Language

Generally, it can be improved

Author Response

The response has been attached as a Word file.

Reviewer 2 Report

Comments and Suggestions for Authors

The article "Genetic Insights Into the Extremely Dwarf Hibiscus syriacus var. micranthus: Complete Chloroplast Genome Analysis and Development of a Novel dCAPS Marker" presents a detailed study on the genetic characteristics of the dwarf variety of Hibiscus syriacus var. micranthus (HSVM), emphasizing its evolutionary significance and ornamental value. The researchers successfully assembled the complete chloroplast genome of HSVM and compared it with those of other Hibiscus species, identifying a unique insertion in the trnL-UAA region exclusive to HSVM. This discovery led to the development of a novel dCAPS marker for distinguishing HSVM and its derived cultivars from other Hibiscus cultivars. The effectiveness of this marker was validated across multiple cultivars. Additionally, phylogenetic analysis illustrated HSVM's close genetic relationship with other woody Hibiscus species, suggesting shared ancestry despite phenotypic differences.

Major issues:

  1. The phylogenetic analyses: there’s no mention of evolutionary model selection and application process to ML analysis. Moreover, the Figure 5 lacks bootstrap support values, and this parameter isn’t mentioned in the text description of tree topology too. I would encourage authors to present the phylogram as a main figure since it’s generally more informative.

  2. Lack of pan-plastomic variation analysis: some variation is described in text, but is focused on selected regions of pairwise identities. Some visualization of variation across chloroplast genome would improve clarity of the results

  3. The manuscript didn’t fully use obtained resources in the term of molecular identification - how many cultivar and species specific molecular diagnostic characters were identified across the palstome and how are they distributed?

Minor issues:

  1. I would recommend to move Tale 1 do supplementary data, since most of the columns have the same values

  2. Why so low coverage SNP (>5 reads) were identified since plastome coverages were high (>2400x). It’s usually fine to keep SNP with frequency > 0.05.

Author Response

(The authors gave the same response as above.)

Reviewer 3 Report

Comments and Suggestions for Authors

The paper deals with the chloroplast genome and the origin of the dwarf variety of Hibiscus syriacus var. micranthus.

The methodology used is correct. The methods are correctly described. The language of the work is clear and understandable.

The tables and figures are correct.

Research problem:

The authors emphasize the importance of research on the dwarf variety of Hibiscus syriacus var. micranthus. And I understand the importance of this variety in horticulture. However, I don't really understand (the authors didn't convince me) what their research means. I don't know what the chloroplast genome study would help with. Isn't it better to analyze the nuclear genome in order to understand why this cultivar is dwarfed?

Moreover, there is already a very good paper on the chloroplast genome of this species (https://www.frontiersin.org/journals/plant-science/articles/10.3389/fpls.2023.1111968/full).

I would ask the authors to try harder to define the novelty of the study to make it clear to the reader, maybe additional results would make the work more interesting?

As for Figure 5. phylogenetic analysis, it shows that H. sinosyriacus should be included in genus H.syriacus. Could the authors elaborate on this in the discussion

Author Response

(The authors gave the same response as above.)

Reviewer 4 Report

Comments and Suggestions for Authors

The manuscript entitled "Genetic Insights into the Extremely Dwarf Hibiscus syriacus var. micranthus: Complete Chloroplast Genome Analysis and Development of a novel dCAPS Marker " is written well. The authors have assembled the complete cp genomes of HSVM, the renowned HS natural monument no. 521, and the HS 'Tamra' cultivar. They performed phylogenetic analysis to determine the evolutionary position of HSVM. Finally, the authors identified intraspecific cp variations in HSVM, leading to the development of a specialized derived cleaved amplified polymorphic sequence marker and its application to cultivar differentiation. The study sheds light on HS's genetic diversity and evolutionary history through the lens of the HSVM cp genome.

Remarks:

Table 1 - "Summary of complete chloroplast genomes of three Hibiscus syriacus specimens." but there are four.

The authors claimed that they sequenced three CP genomes but provided data just for one (Figure 2 and Table 2).

Figure 8: The marker used is not proper for the visualized fragments. The authors said (lines 336-337), "...showed cleaved fragments of 105 bp and 28 bp. In contrast, the PCR products from the remaining HS cultivars were not cleaved, resulting in 134 bp bands." It is not visible on the gel the 28 bp fragments.

The authors used sequences with GenBank accession numbers for Phylogenetic analysis but did not refer to which database. I found someone in the NCBI database, but some are missing there!

Line 152: RT-PCR ??? Which analysis precisely what the authors mean?

Author Response

(The authors gave the same response as above.)

Round 2

Reviewer 2 Report

Comments and Suggestions for Authors

Dear Authors,

Thank you for addressing  some of issues in the revised version, but there is still one not resolved or answered:

Q:
The manuscript didn’t fully use obtained resources in the term of molecular identification -how many cultivar and species specific molecular diagnostic characters were identified across the palstome and how are they distributed?
A:
In Korea, approximately 150 varieties of Hibiscus syriacus, commonly known as the Korean rose of sharon, have been developed. Most of these are preserved within the ‘Hibiscus clonal archive’ at the National Institute of Forest Science. Among them, the variant HSVM is preserved, characterized by its white flowers with red eye spots and narrow petals. In this study, we included HSVM, ten cultivars developed using it, 26 similar cultivars, and six cultivars with significantly different phenotypes, incorporating most of the cultivars that have
even a small potential for dispute in identification. Finding Intra-specific variations in cp genome is extremely difficult. In particular, when
compared with other major Korea rose of sharon cultivars assembled together, HSVM had only one unique intraspecific variation. To aid understanding of the distribution of variations, we used the mVISTA program to visualize the overall distribution of variations and included this in our study

I agree that intraspecific variation could be low or even absent in plastomes, but the question was about interespecific diagnostic nucleotides too. So how many species specific molecular diagnostic characters were identified across the plastome and how are they distributed across chloroplast genome?

Author Response

(The authors gave the same response as above.)

Reviewer 3 Report

Comments and Suggestions for Authors

Dear Soon-Ho Kwon,

Thank you for the substitutions made in the article and the substantive explanations to my questions. After these changes, in my opinion, the article is suitable for publication.

Author Response

Dear reviewer, 

We appreciate your valuable feedback. Despite your positive response, we have further improved the English quality of our manuscript.

Best regards.

Soon-Ho Kwon

Reviewer 4 Report

Comments and Suggestions for Authors

This study illuminates HS's genetic diversity and evolutionary narrative through the cp genome analysis of the HSVM and Korea-native HS. HSVM is validated as a valuable genetic resource for genetic research and breeding. This research, which discovered the unique role of the P8 region within the cp genome, highlights the importance of intronic sequences in plant adaptation and diversity, providing a direction for future research focused on intronic variations within the cp genome and their direct role in plant phenotypic expression. After the changes and clarifications made, as a result of the two reviewers, the article entitled: "Genetic Insights into the Extremely Dwarf Hibiscus syriacus var. micranthus: Complete Chloroplast Genome Analysis and Development of a novel dCAPS Marker" now appears to be a version suitable for publication in the journal Current Issues in Molecular Biology.

Author Response

(The authors gave the same response as above.)
